# Multivariate Analysis for Assessing Irrigation Water Quality: A Case Study of the Bahr Mouise Canal, Eastern Nile Delta

**Mohamed K. Abdel-Fattah [1]**, **Sameh Kotb Abd-Elmabod [2,3,\***], **Ali A. Aldosari [4]**, **Ahmed S. Elrys [1]** and **Elsayed Said Mohamed [5,\***]

[1] Soil Science Department, Faculty of Agriculture, Zagazig University, Zagazig 44519, Egypt; mkabdelfattah@zu.edu.eg (M.K.A.-F.); Aselrys@zu.edu.eg (A.S.E.)
[2] Soils and Water Use Department, Agricultural and Biological Research Division, National Research Centre, Cairo 1262, Egypt
[3] MED Soil Research Group, Department of Crystallography, Mineralogy and Agricultural Chemistry, Seville University, 41012 Seville, Spain
[4] Geography Department, King Saud University, 11451 Riyadh, Saudi Arabia; adosari@ksu.edu.sa
[5] National Authority for Remote Sensing and Space Sciences, Cairo 11843, Egypt
[\*] Correspondence: sk.abd-elmabod@nrc.sci.eg (S.K.A.-E.); elsayed.salama@narss.sci.eg (E.S.M.)

**Abstract:** Water scarcity and suitable irrigation water management in arid regions represent tangible challenges for sustainable agriculture. The current study aimed to apply multivariate analysis and to develop a simplified water quality assessment using principal component analysis (PCA) and the agglomerative hierarchical clustering (AHC) technique to assess the water quality of the Bahr Mouise canal in El-Sharkia Governorate, Egypt. The proposed methods depended on the monitored water chemical composition (e.g., pH, water electrical conductivity (ECiw), $Ca^{2+}$, $Mg^{2+}$, $Na^+$, $K^+$, $HCO_3^-$, $Cl^-$, and $SO_4^{2-}$) during 2019. Based on the supervised classification of satellite images (Landsat 8 Operational Land Imager (OLI)), the distinguished land use/land cover types around the Bahr Mouise canal were agriculture, urban, and water bodies, while the dominating land use was agriculture. The water quality of the Bahr Mouise canal was classified into two classes based on the application of the irrigation water quality index (IWQI), while the water quality was classified into three classes using the PCA and AHC methods. Temporal variations in water quality were investigated, where the water qualities in winter, autumn, and spring (January, February, March, April, November, and December) were classified as class I (no restrictions) based on IWQI application, and the water salinity, sodicity, and/or alkalinity did not represent limiting factors for irrigation water quality. On the other hand, in the summer season (May, June, July, August, and October), the irrigation water was classified as class II (low restrictions); therefore, irrigation processes during summer may lead to an increase in the alkalinity hazard. The PCA classifications were compared with the IWQI results; the PCA classifications had similar assessment results during the year, except in September, while the water quality was assigned to class II using the PCA method and class I by applying the IWQI. Furthermore, the normalized difference vegetation index (NDVI) around the Bahr Mouise canal over eight months and climatic data assisted in explaining the fluctuations in water quality during 2019 as a result of changing the crop season and agriculture management. Assessments of water quality help to conserve soil, reduce degradation risk, and support decision makers in order to obtain sustainable agriculture, especially under water irrigation scarcity and the limited agricultural land in such an arid region.

**Keywords:** water quality criteria; IWQI; arid region; PCA; land use

## 1. Introduction

Surface water quality is a very sensitive and global environmental issue that is important for long-term economic development and environmental sustainability [1–3]. Awareness and attention to water irrigation quality have increased worldwide in recent years, and new approaches have been developed to achieve the sustainable management of water resources [4,5]. In the same context, the shortage of water resources has become a big problem in many countries, particularly under continued population growth, accelerated industrialization, rapid urbanization, and global climate change [6,7]. Therefore, water scarcity and sustainable irrigation water management have become global challenges for sustainable agriculture development in order to produce sufficient food to satisfy the population's food requirements [8–10].

Agricultural production in Egypt is largely dependent on the River Nile, which contributes 73% of the total irrigation requirement (55 billion $m^3$ per year) [11–13]. Egyptian agricultural land represents only 3.8% ($km^2$) of the total area (1.01 million $km^2$), and the Nile delta provides 65% (24,700 $km^2$) of the total agricultural land [13,14]. Therefore, Egypt faces great challenges due to arid climate, fixed Nile water share, limited agricultural land, and rapid population growth. The water renewable resources per capita have dropped intensely to 700 $m^3$/capita, which moved Egypt below the water poverty level [15]. In this context, the amount of water shares per capita in Egypt may decrease to 500 $m^3$/capita by 2025 [16]. Therefore, the limited and fixed water resources, as well as ever-increasing water demands, are the main issues in accelerating the practice of agricultural drainage water reuse as an alternative resource to fill the gap between water supply and demand. The Egyptian national water strategy states that the practice of drainage water reuse would fulfil irrigation water demands for newly reclaimed areas in the Eastern and Western Nile Delta regions [15,16].

Irrigation water contains several dissolved salts [17–19]. The characteristics and amount of these dissolved salts depend on the water source and its chemical composition. The most ordinarily dissolved ions in water are calcium ($Ca^{2+}$), sodium ($Na^+$), magnesium ($Mg^{2+}$), sulfate ($SO_4^{2-}$), nitrate ($NO_3^-$), chloride ($Cl^-$), boron (Br), carbonate ($CO_3^{2-}$), and bicarbonates ($HCO_3^-$). The proportion and concentration of these dissolved ions are used to determine the suitability of water for irrigation [20–22]. Water irrigation quality for agricultural use is determined based on its impact on crop yield (quality and quantity), as well as its impact on soil physiochemical properties [23]. Most soil problems (e.g., salinity, sodicity, contamination, and restricted infiltration) are due to the use of low-quality water for irrigation [24].

Nile river water is characterized by its high quality from the upper Nile countries to the mouth of the river in Egypt. The annual average electrical conductivity (EC) is about 0.15 dS $m^{-1}$ in the Victoria and Atbara Lakes, while the EC increases to reach 0.70 dS $m^{-1}$ in the Rosetta branch north of the Nile delta. Accordingly, the EC values of the river Nile range from 0.27 to 0.46 dS $m^{-1}$ in upper Egypt [25]. In addition, calcium and magnesium are the dominant cations in the water of the Nile river in upper Egypt, while sodium and potassium are the dominant cations in the Nile water of lower Egypt [26]. The Bahr Mouise canal represents the main canal for irrigation and municipal water in the El-Sharkia Governorate, Egypt [8]. The contamination of the Bahr Mouise canal is due to the brick industry in close areas (Menia El-Kamh and Hehia), soup, and oil industries in El-Zagazig city, as well as human settlement effluent [27]. Generally, the use of low-quality water causes several environmental impacts on soil, plants, animals, and humans [28,29]. Urbanization and human activities impact almost all freshwater bodies [30].

The low quality of irrigation water is characterized by a reduction of dissolved oxygen, a lower transparency, a high electric conductivity, a high alkalinity, a water temperature increase, and high levels of total dissolved solids [31]. The irrigation water quality index (IWQI) represent a gathering of individual water parameters that are expressed in a single numerical expression in order to judge the use of water for irrigation purposes [32].

Many researchers have used remote sensing and geographic information system (GIS) techniques to study water quality and environmental contamination to find the relationship between water

pollution and different land uses under diverse environmental conditions [33,34]. Different studies have proven the negative impacts of urban and industrial areas on water quality [35,36]. Land cover/land use (LCLU) changes are related to human activity, while rapid urbanization represents a real threat to the most fertile soils and water quality in the Nile delta [37]. Therefore, creating LCLU maps helps in monitoring the changes of land cover, and correlating the variation in water quality with LCLU changes helps to obtain optimal solutions and to improve irrigation water quality [38–42]. The spatiotemporal data provided by remote sensing analyses play a vital role in observing and investigating land cover changes over time [43]. The normalized difference vegetation index (NDVI) has been used for monitoring vegetation [44,45], calculating crop cover [46,47], monitoring drought [48], and assessing agricultural drought at the national level [49].

Principal component analysis (PCA) can convert multiple parameters into comprehensive indicators based on dimensional reduction and can decrease the calculation complexity of using many variables in a conventional statistical analysis [50–54]. In pervious study, various multivariate statistical analyses were applied to assess the river water quality based on using 13 parameters, and the cluster analysis (CA) method was used to group the 12 months of monitoring into three periods, as well as to group 18 sampling sites into three groups [55]. The discriminant analysis (DA) aimed to identify the significant parameters among the temporal and spatial groups, and the PCA method aimed to standardize water quality, examine the differences between groups, identify the main sources of contamination. Most former studies have used the PCA method to assess the water of rivers and lakes for irrigation proposes [56,57]. On the other hand, many indices have been used for monitoring recycled wastewater while considering wastewater reclamation standards [58–61].

The study objective was to assess the irrigation water quality of the Bahr Mouise canal at six locations over the 12 months of 2019 using a multivariate analysis (i.e., the IWQI and the development of a new classification method that depends on PCA and agglomerative hierarchical clustering (AHC) technique methods). Additionally, the study aimed to link the changes in the NDVI, LCLU, water use, and climate conditions with the fluctuations in water quality classifications in each month and site.

## 2. Materials and Methods

### 2.1. Study Area and Sampling

Bahr Mouise is the main irrigation canal located in the El-Sharkia Governorate (Egypt) between a latitude of 30°28′–30°50′ N and a longitude of 31°12′–31° 40′ E (Figure 1). The canal length is about 83 km, and it passes through several villages, towns, and cities (Menia El-Kamh, El-Zagazig, Hehia, and AwladSakr). The average daily discharge of the Bahr Mouise canal ranges between 5.0 and 12.5 million $m^3$, and it provides the irrigation water requirements of 340,000 hectares [27]. To assess the water quality of Bahr Mouise, monthly samples (from January to December, 2019) were collected from six random sampling sites with irregular distances between one point and another along the central part of the canal. At each sampling site, four samples were collected using a portable water sampler, and all water measurements were carried out within 24 h after sampling. Samples were prepared to analyze salinity (measured as electrical conductivity), pH, and major ions (i.e., $Ca^{2+}$, $Mg^{2+}$, $Na^+$, $K^+$, $HCO_3^-$, $Cl^-$, and $SO_4^{2-}$) following the standard methods [55].

### 2.2. Remote Sensing Data and Analysis

The supervised imaging classification of the Bahr Mouise canal basin was carried out using the satellite image of the Operational Land Imager (OLI), which acquired data on 20 January 2019 with a spatial resolution of 30 m. The image preprocessing was done based on the radiometric and atmospheric calibrations using the ENVI software 5.3 (Exelis Visual Information Solutions, Boulder, CO, USA). The satellite image was classified using the supervised maximum likelihood classification to obtain an LCLU map [62]. The classification determined the different and the dominant LCLUs on the adjacent areas to the Bahr Mouise canal. Urban land use might have negative impacts on water

quality [63]. The NDVI is a remote sensing measure that is used to identify crop growth status and earth surface vegetation [47]. Vegetation indices also help to distinguish the distribution of vegetation and soil based on the distinctive reflectance patterns of vegetation and other land surfaces [64,65]. The NDVI can be calculated by Equation (1) [66]:

$$NDVI = \frac{(NIR - RED)}{(NIR + RED)} \tag{1}$$

where NIR is the near-infrared band and RED represents the red band. NDVI values range from −1 to 1.

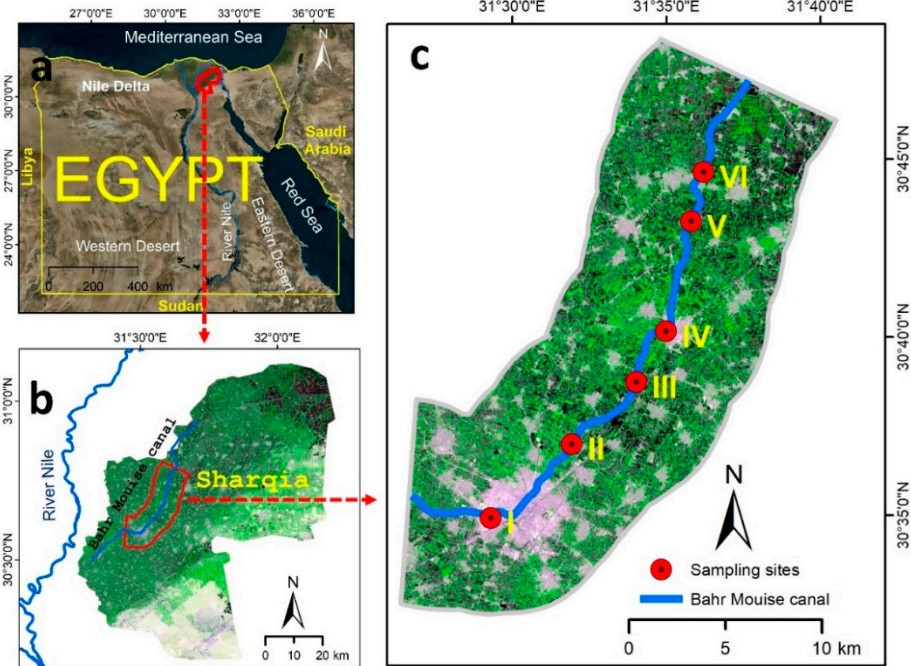

**Figure 1.** Location of the Bahr Mouise canal in the El-Sharkia Governorate, Egypt ((**a**) and (**b**)), as well as the six sampling sites (**c**).

### 2.3. Traditional Irrigation Water Quality Criteria

Various traditional equations have been used to judge the suitability of water for irrigation based on chemical parameters, such as the soluble sodium percentage (SSP) (Equation (2) [67]), the sodium absorption ratio (SAR) (Equation (3) [67]), the magnesium adsorption ratio (MAR) (Equation (4) [68]), the sodium-to-calcium activity ratio (SCAR) (Equation (5) [69], residual sodium carbonate (RSC) (Equation (6) [67]), residual sodium bicarbonate (RSBC) (Equation (7) [29]), the permeability index (PI) (Equation (8) [67,70]), and the Kelly ratio (KR) (Equation (9) [71]). Additionally, ion contents (i.e., $Na^+$, $Cl^-$, $HCO_3^-$, $SO_4^{2-}$, and $NO_3^-$), water electrical conductivity (ECiw), and pH were used in assessing irrigation water quality. The equations used to evaluate the irrigation water quality are explained below, and all ion concentrations are expressed in mmol $L^{-1}$.

$$SSP = \frac{Na^+}{Na^+ + K^+ + Ca^{2+} + Mg^{2+}} \times 100 \tag{2}$$

$$SAR = \frac{Na^+}{\sqrt{\frac{Ca^{2+} + Mg^{2+}}{2}}} \tag{3}$$

$$MAR = \frac{Mg^{2+}}{Ca^{2+} + Mg^{2+}} \times 100 \tag{4}$$

$$SCAR = \frac{Na^+}{\sqrt{Ca^{2+}}} \tag{5}$$

$$RSC = \left(HCO_3^- + CO_3^{2-}\right) - \left(Ca^{2+} + Mg^{2+}\right) \tag{6}$$

$$RSBC = HCO_3^- - Ca^{2+} \tag{7}$$

$$PI = \frac{Na^+ + \sqrt{HCO_3^-}}{Na^+ + Ca^{2+} + Mg^{2+}} \times 100 \tag{8}$$

$$KR = \frac{Na^+}{Ca^{2+}} \tag{9}$$

*2.4. Irrigation Water Quality Index*

The obtained water parameters of the Bahr Mouise canal, such as ECiw, $Na^+$, SAR, $HCO_3^-$, and $Cl^-$, were used in applying the IWQI that was developed by [72], as explained in Equation (10);

$$IWQI = \sum_{i=1}^{n} q_i w_i \tag{10}$$

where $q_i$ and $w_i$ refer to quality measurement values and aggregation weights, respectively. The $q_i$ value was calculated based on Equation (11):

$$q_i = q_{imax} - \left(\frac{\left(x_{ji} - x_{inf}\right) \times q_{imap}}{x_{imap}}\right) \tag{11}$$

where $q_{max}$ is the maximum value of qi for each class, $x_{ij}$ is the observed value for parameter, $x_{inf}$ is the lower limit of the class in which the parameter belongs, $q_{iamp}$ is class amplitude, and $x_{amp}$ is class amplitude to which the parameter belongs. Regarding the $w_i$ (aggregation weights), each parameter weight used in the IWQI was obtained via the methods of Meireles et al. [72]. The parameter weights were 0.211, 0.204, 0.202, 0.194, and 0.189 for ECiw, $Na^+$, $HCO_3^-$, $Cl^-$, and the SAR, respectively. The IWQI is a dimensionless parameter ranging from 0 to 100. Following [72], irrigation water quality was classified into four classes based on IWQI value: null (85–100), low (70–85), moderate (55–70), high (40–55), and severe restriction (0–40).

*2.5. Principal Component Analysis*

PCA was used to assess the irrigation water quality. Before applying the PCA analysis, the variable normality was checked using the Shapiro–Wilk test, and the correlation between different variables was measured by the Pearson correlation. Bartlett's sphericity and Kaiser–Meyer–Olkin (KMO) tests were done to verify the data independency [73] before the PCA if the value of the KOM test was greater than 0.5, indicating the adequacy of performing the PCA analysis [74,75] (Table S1). Then, the PCA was performed using SPSS Software version 25 (SPSS Inc., Chicago, IL, USA) to obtain the principal components (PCs) that had high correlation with the studied variables. According to [76], a principal component can be explained using Equations (12) and (13):

$$Z_k = ak_1 X1 + ak_2 X_2 + \ldots \ldots + ak_n X_n \tag{12}$$

$$Z = (z_1 V_1 + z_2 V_2 + \ldots \ldots + z_n V_n)/(V_1 + V_2 + \ldots \ldots + V_m) \tag{13}$$

where z is the component score, a is the component loading, X is the measured value of a parameter, k is the component number, n is the total number of parameters, Z is the comprehensive score, V is the total variance of each component, and m is the total number of components.

The number of PCs was selected depending on the eigenvalue. The PCs that had an eigenvalue greater than one were kept, and the rest were removed, and 70% or greater of the total variability had to be expressed by the selected PCs. Subsequently, the correlations between the selected PCs and the observed variables could be explained with factor loading. The factor loadings could be estimated based on Equation (14):

$$\text{Factor loadings} = \text{Eigenvectors} \times \sqrt{\text{Eigenvalue}} \tag{14}$$

A factor loading of >0.3 is considered to be significant, >0.4 is more significant, and >0.50 is very significant [77]. The factor scores represent PC contributions to explain each variable's variance. Each item's contribution to the PC score depends on how strongly it relates to the PC.

Classification Method

Based on the principal component scores of each month, AHC, one of the most popular clustering methods, was done using Ward's method in order to obtain the water quality classification. A small value of the sum of squared error (SSE) illustrates that all instances in the cluster are near to the cluster mean and thus have high degrees of similarity. The two clusters that minimally increase the value of the SSE were joined by Ward's method. The SSE is a measure of the distance between a cluster's instance attribute and mean attribute values, according to Equations (15) and (16):

$$\text{SSE} = \sum_i^{Sx} \sum_j^{Sy} \left| Xij - Yi \right|^2 \tag{15}$$

$$D(x,y) = \text{SSE}(xy) - (\text{SSE}(X) + \text{SSE}(Y)) \tag{16}$$

where Sx is the number of instances, Sy is the number of attributes, Xij is the value of attribute j in instance i, Yi is the mean value of attribute j, D (x,y) is the SSE change after joining clusters x and y, SSE(xy) is the SSE of joined clusters x and y, SSE(X) is the SSE of cluster x, and SSE(Y) is the SSE of cluster y.

Lastly, the dendrogram is a part of AHC outputs that shows the progressive data grouping; thus, it helps to group data in a suitable number of classes based on dissimilarity. The distances between the class centroids, as well as distances between the central objects, were produced using the Euclidean distance method. The Euclidean distance between two instances with k attributes is calculated using Equation (17):

$$d_{(x,y)} = \sqrt{\sum_{k=1}^{n} (x_k - y_k)^2} \tag{17}$$

where $d_{(x,y)}$ distance between the two instances, n is the number of dimensions (attributes), and $x_k$ and $y_k$ are, respectively the k attributes (components) or data objects x and y.

## 3. Results

### 3.1. Land Use/Cover (LCLU)

Figure 2 shows the LCLU classes around the Bahr Mouise canal, where four classes were observed (agriculture, urban, water bodies, and bare land) that generally represent the main classes in the Nile Delta [78,79]. Agriculture (vegetation) land use is the dominant class, and urban (residential buildings and industrial areas) is the second dominant class, as it represents about 25% of the total studied area. Figure 2 shows that the Bahr Mouise canal is directed from the south towards the north. The water of

the canal is affected by contamination sources from agriculture drainage and human effluent from urban areas. Figure S1 illustrates the impact of different land uses on the irrigation water quality of the Bahr Mouise canal, and the variations in variable concentrations (i.e., pH, ECiw, $Ca^{2+}$, $Mg^{2+}$, $Na^+$, $K^+$, $HCO_3^-$, $Cl^-$, and $SO_4^{2-}$), while the high concentration values were observed in the areas that had mixed urban and agricultural LCLU, which may negatively impact the water quality of this site location.

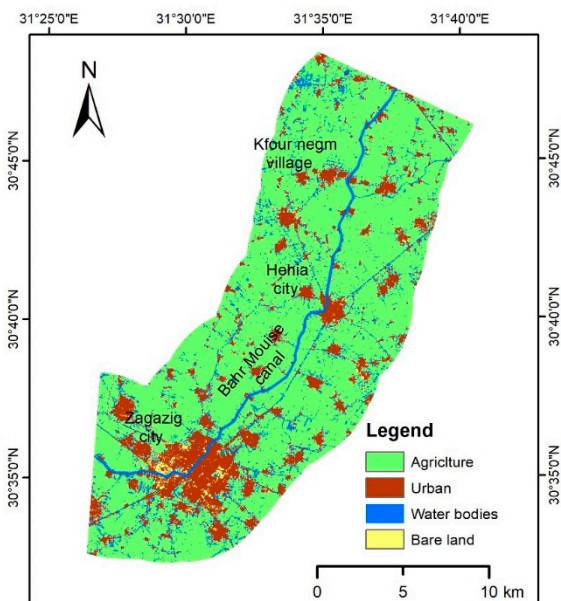

**Figure 2.** The main land cover land use (LCLU) classes around the Bahr Mouise canal.

Figure 3 illustrates the values of the NDVI for water bodies that had the lowest values (from −0.1 to 0.0), bare land, and urban-occupied low values (0.0–0.1), while the higher NDVI values were assigned to the cropland where the values ranges from 0.1 to 0.6.

Figure 3 demonstrates the changes in the NDVI values over time (eight months during 2019). In May, farmers were preparing farmland (sowing and planting the summer crops); thus, the majority of the agricultural land had low NDVI values compared with the NDVI values in August and September where the summer crops were fully grown. Following the same trend, the winter crops were cultivated from October and November, and till January, the crops were still in the primary growth stage. As such, the NDVI values were low in January. On the contrary, the values were increased in February and March as a result of the growth stage development of the winter crops.

*3.2. Climate and Water Use*

The monthly climate parameters (i.e., mean temperature (Tmean), precipitation, and the annual potential evapotranspiration ($ET_0$)) of the representative meteorological station of the El-Sharkia Governorate are presented graphically in Figure 4A. The mean annual temperature reached 21.3 °C, with an annual rainfall of 57 mm. In the same context, the $ET_0$ had a value of 1045 mm, and according to the aridity index, the twelve months of the study period had arid climate conditions (in which the $ET_0$ exceeded the actual precipitation) [11].

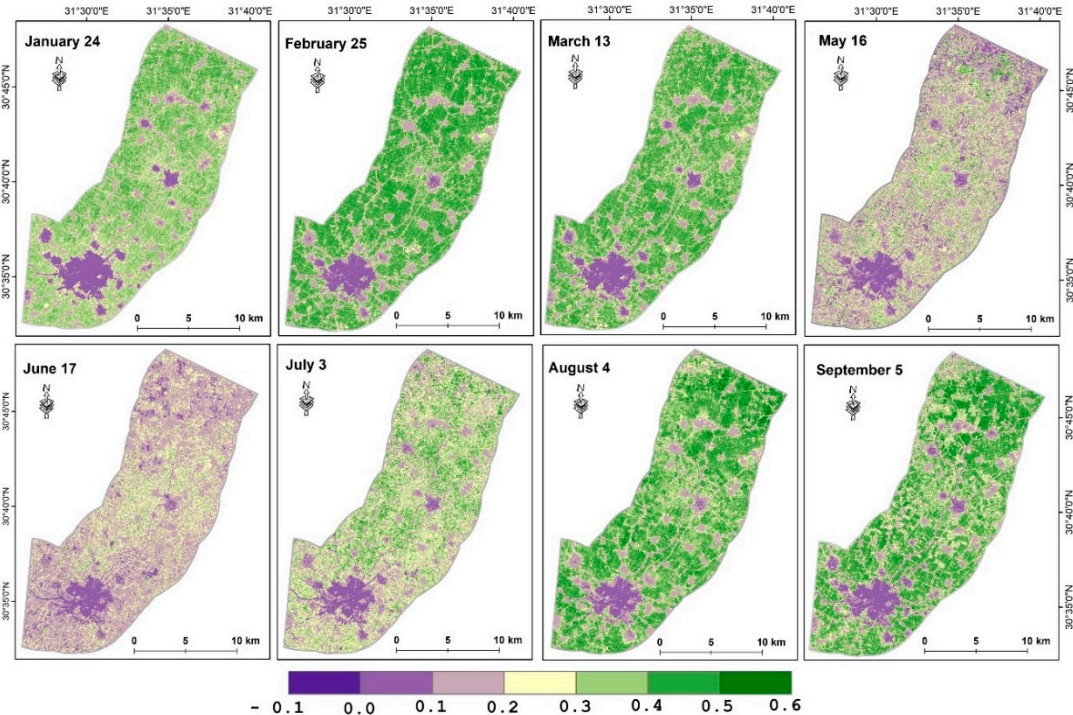

**Figure 3.** The variation in the normalized difference vegetation index (NDVI) over eight months during 2019.

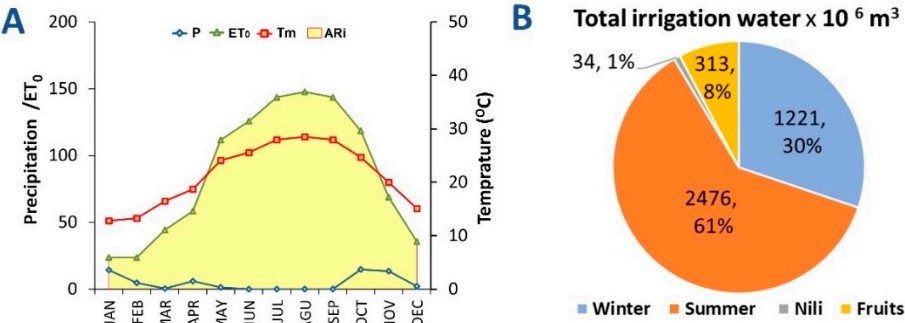

**Figure 4.** Volumetric climate graph of the representative metrological station for the studied area (**A**). Governorate by crop season (**B**). Tm: mean temperature in °C; $ET_0$: potential evapotranspiration in mm; P: precipitation in mm; and Ari: aridity index. Pie chart shows the distribution of water irrigation quantity in the El-Sharkia.

The agricultural water requirements in the El-Sharkia Governorate for summer crops (cultivated from February to May), winter crops (cultivated from September to November), Nili crops (cultivated from July to August), and fruits were 2.47, 1.22, 0.31, and 0.03 billion $m^3$, respectively [11]. The major summer crops are maize, rice, sorghum, cotton, sunflower, sesame, sugarcane, soybean, and onions. The major winter crops are beans, wheat, sugar beet, barley, onion, alfalfa, garlic, and lupine. The Nili crops are sunflower, maize, rice, sorghum, and onions. Consequently, the summer crops had the greatest water irrigation requirements (Figure 5B).

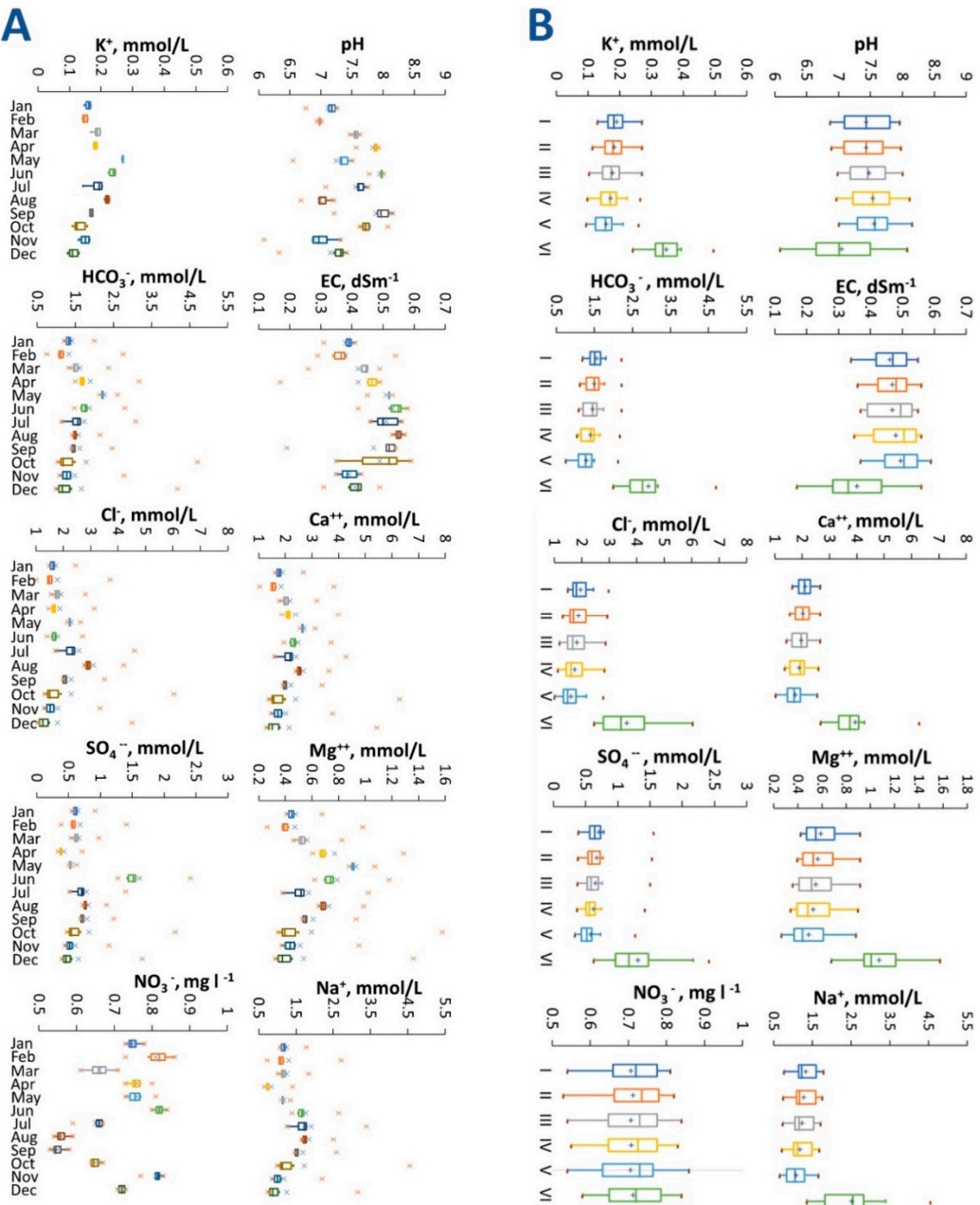

**Figure 5.** Spatiotemporal analysis of the chemical water composition—monthly (**A**) and at each sampling site (**B**).

### 3.3. Validity of Bahr Mouise Water based on Traditional Criteria Analysis

Water pH: According to the Food and Agriculture Organization (FAO) [80], the acceptable level of pH for irrigation water is 6.0–8.5. The pH of the Bahr Mouise canal water was within the recommended ranges, varying from 6.89 to 7.95 over the twelve months with an average of 7.41 ± 0.39 (Figure 5 and Table S2).

Salinity hazard: The United States Department of Agriculture (USDA) [67] classified the irrigation water based on ECiw into four classes: low salinity (C1: <0.25 dS m$^{-1}$), medium salinity (C2 ranged from 0.25 to 0.75 dS m$^{-1}$), high salinity (C3 ranged from 0.75 to 2.25 dS m$^{-1}$), and very high salinity (C4: >2.25 dS m$^{-1}$). Therefore, the EC of the Bahr Mouise canal water was assigned to the medium salinity class where the values of ECiw ranged from 0.38 to 0.55 dS m$^{-1}$ with an average of 0.46 dS m$^{-1}$ (Figure 5 and Table S2).

Another classification of irrigation water based on ECiw was provided by Gupta [69]: low salinity (C-1: ECiw 0.2–1.5 dS m$^{-1}$), medium salinity (C-2: ECiw 1.5–3.0 dS m$^{-1}$), high salinity (C-3: ECiw 3.0–5.0 dS m$^{-1}$), and very high salinity (C-4: ECiw 5–10 dS m$^{-1}$). Accordingly, the EC of the Bahr Mouise canal water was assigned to the low salinity class. Moreover, the ECiw values were converted to total dissolved solid (TDS) values based on the equation TSD, mg L$^{-1}$ = EC, dS m$^{-1}$ × 640 that was expressed by the authors of [69]; the TDS ranged between 243 and 352 mg L$^{-1}$ with an average of 294 mg L$^{-1}$.

Sodicity hazard: The most important indicators of sodicity water irrigation hazard are the SSP (Equation (2)) and the SAR (Equation (3)). When there is a high concentration of sodium ions in irrigation water, Na$^+$ ions tend to be absorbed by clay particles, displacing Mg$^{++}$ and other cations. Water with an SSP greater than 60% breaks down the soil's physical properties, which may occur due to sodium accumulation [81]. The calculated SSP values varied from 19.96% to 37.32% with a mean value of 31.40%, indicating a low restriction degree on using the water for irrigation. Kumar et al. [82] note that water with an SSP < 50% is good quality and suitable for irrigation purposes.

The USDA [67] classifies irrigation water based on the SAR into four classes: low (S1: SAR <10), medium (S2: SAR 10–18), high (S3: SAR 18–26), and very high (S4: SAR >26). Thus, the SAR values of the Bahr Mouise canal were in the low class and ranged from 0.67 to 1.56 mmol L$^{-1}$ with an average of 1.19 mmol L$^{-1}$ (Figure 6 and Table S3).

Another classification of irrigation water based on the combination of EC and the SAR is provided by the USDA [67]. The classes of this classification are: very good water (C1S1), good water (C1S2, C2S2, and C2S1), usable water (C1S3, C2S3, C3S3, C3S2, and C3S1), and usable water with caution (C1S4, C2S4, C3S4, C4S4, C4S3, C4S2, and C4S1). The water of the Bahr Mouise canal was assigned to the good water C2SI class (Figure 6 and Table S3).

Regarding the SAR/SCAR ratio (the SCAR was calculated as shown in Equation (5)), the irrigation water was classified into six classes of sodicity [83]: non-sodic water (S-0: SAR/SCAR < 5), normal water (S-1: SAR/SCAR 5–10), low sodicity water (S-2: SAR/SCAR 10–20), medium sodicity water (S-3: SAR/SCAR 20–30), high sodicity water (S-4: SAR/SCAR 30–40), and very high sodicity water (S-5: SAR/SCAR > 40). Accordingly, the water of the Bahr Mouise canal was found to fall in the non-sodic water (S-0) class, where the values of the SAR/SCAR ratio ranged from 1.22 to 1.27 with an average of 1.25 (Figure 6 and Table S3).

Alkalinity hazard: According to the USDA [67], irrigation water is classified based on RSC (Equation (6)) into three classes: 1—safe for irrigation (RSC < 1.25); 2—medium hazard (RSC ranged from 1.25 to 2.5); and 3—extreme hazard (RSC > 2.5). Thus, the water of the Bahr Mouise canal was found to be safe for irrigation, and the values of RSC ranged between −1.83 and −0.96 mmol L$^{-1}$ with an average of −1.22 mmol L$^{-1}$ (Figure 6 and Table S3).

Moreover, using the RSC/RSBC ratio (the RSBC was calculated as shown in Equation (7)), the irrigation waters could be classified into six classes of alkalinity [83]. The six classes of the RSC/RSBC ratio are non-alkaline water (A-0: RSC/RSBC equals a negative value), normal water (A-1: RSC/RSBC equals zero), low alkalinity water (A-2: RSC/RSBC equals 2.5), medium alkalinity water (A-3: RSC/RSBC ranges from 2.5 to 5), high alkalinity water (A-4: RSC/RSBC ranges from 5 to 10), and very high alkalinity water (A-5: RSC/RSBC > 10). Accordingly, 83.33% of the studied water samples were assigned to the A-2 class, 16.67% were allocated to the A-3 class, and the values of the RSC/RSBC ratio were ranged between 1.66 and 3.09 with an average of 2.12 (Figure 6 and Table S3).

Permeability index: The PI value is used as an indicator of irrigation water suitability (Equation (8)). The PI can be classified into three classes: excellent (class I: >75%), good (class II: 25–75%), and unsuitable (class III: <25%) [84]. The PI values of the Bahr Mouise canal ranged between 55.48 and 67.91, so the water of the Bahr Mouise canal was in the good water class for irrigation.

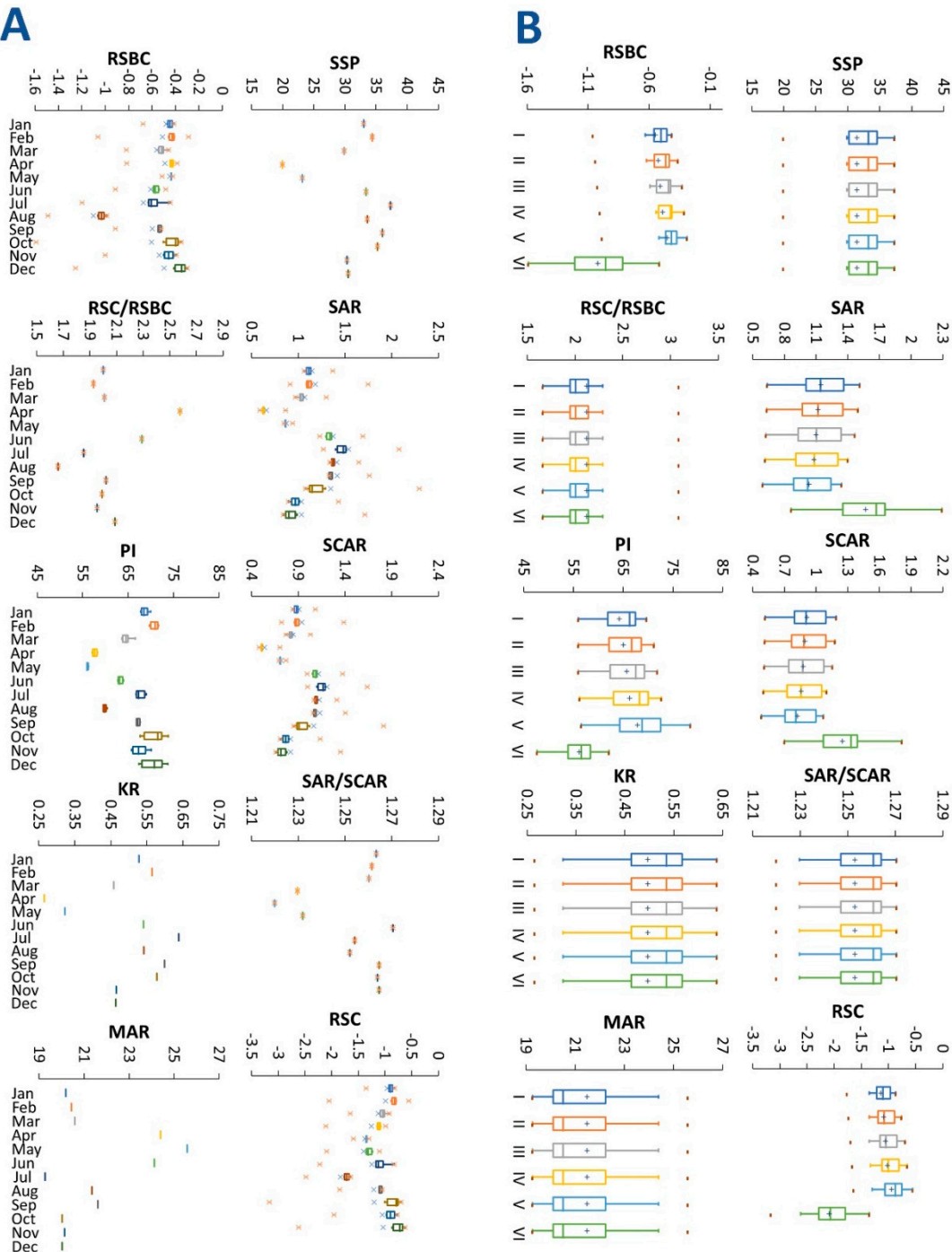

**Figure 6.** Spatiotemporal assessment of the Bahr Mouise canal water using traditional criteria—monthly (**A**) and spatially in each sampling site (**B**). RSBC: residual sodium bicarbonate; SSP: soluble sodium percentage; SAR: sodium absorption ratio; PI: permeability index; SCAR: sodium-to-calcium activity ratio; KR: Kelly ratio; MAR: magnesium absorption ratio; and RSC: residual sodium carbonate.

Magnesium ratio: The magnesium hazard of irrigation water is evaluated using the MAR [85]. The MAR can be classified into two classes: safe water (Class I: <50) and water with $Mg^{2+}$ hazard (Class II: >50). The water of the Bahr Mouise canal is safe for irrigation as the values of MR ranged from 19.28% to 25.59% with an average value of 21.48% (Figure 6).

Kelly ratio: The KR is calculated based on the measurement of sodium against calcium and magnesium (Equation (9)). If the KR > 1, this indicates an excess of sodium in the water [71].

Therefore, water with a KR < 1 is suitable for irrigation, while a greater ratio (KR > 1) is unsuitable [86]. The KR values of the Bahr Mouise canal water ranged from 0.27 to 0.64 with an average value of 0.50; consequently, the water is suitable for irrigation (Figure 6).

Specific ion toxicity: Sodium ($Na^+$), chloride ($Cl^-$), bicarbonate ($HCO_3^-$), and nitrate ($NO_3^-$) ions are considered to be the most common toxic ions in irrigation water. According to guidelines of water quality for irrigation presented by the FAO [80], the $Na^+$, $Cl^-$, and $NO_3^-$ ions of the Bahr Mouise canal were found to be at levels characterized as safe for irrigation purposes. The values were $Na^+$ < 3 (expressed as the SAR), $Cl^-$ < 3, and $NO_3^-$ < 5 mg $L^{-1}$. On the other hand, the $HCO_3$ concentrations ranged between slight and moderate concentrations, from 1.33 to 2.26 mmol $L^{-1}$ with an average of 1.69 mmol $L^{-1}$ (Figure 6 and Table S3).

### 3.4. Irrigation Water Quality Index (IWQI)

The IWQI evaluation results of the Bahr Mouise canal indicated that the water quality was assigned to class I—"No restriction"—for January, February, March, April, September, November, and December, when the IWQI values ranged between 85 and 100 (Figure 7A). Therefore, the water can be used in almost types of soil and crops, and it cannot cause any long-term salinity/sodicity problems [72]. The water quality in May, June, July, August, and October was assigned to class II—"Low restriction"—(Figure 7A). Consequently, the water can be safely used to irrigate soils with coarse and medium textures (high contents of sand and silt) and a with moderate permeability status.

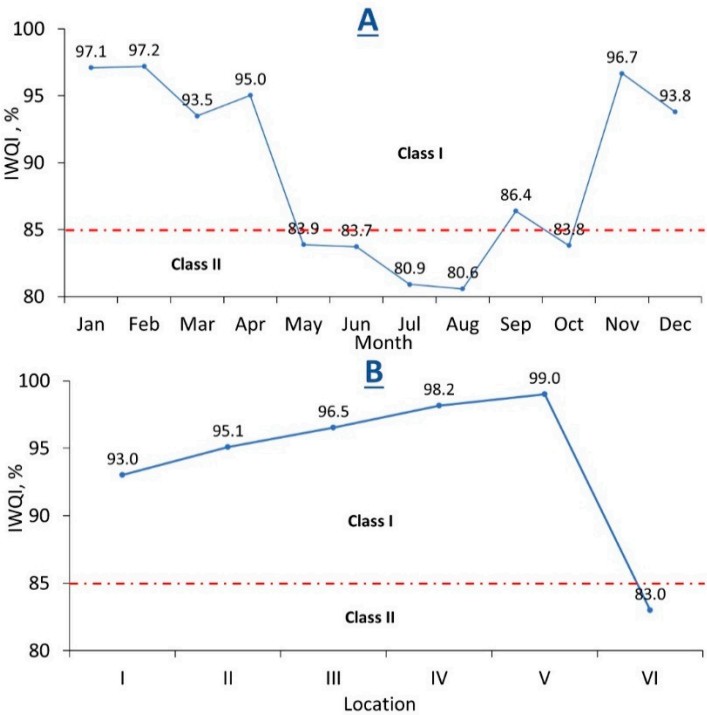

**Figure 7.** Temporal (**A**) and spatial variations (**B**) in the irrigation water quality index (IWQI) of the Bahr the Mouise canal.

On the other hand, to avoid soil sodicity in heavy textures (high clay content) due to irrigation with low quality water, soil leaching is recommended because the water may cause an elevated risk to salinity-sensitive plants. Only sampling site VI was classified as class II—the rest of the studied sites were classified as class I (Figure 7B).

*3.5. Principal Component Analysis (PCA)*

### 3.5.1. Correlation Between the Chemical Constituents of Irrigation Water

The normality of the data was checked using the Shapiro–Wilk test, which indicated that most of the studied variables, except for $Cl^-$ and $SO_4^{2-}$, followed a normal distribution (Table S2). The correlation matrix between all variables of the Bahr Mouise canal water is presented in Table S4. The correlation matrix shows a positive correlation between ECiw and $Ca^{2+}$ (0.90), $Mg^{2+}$ (0.68), $Na^+$ (0.72), $K^+$ (0.77), $HCO_3^-$ (0.57), $Cl^-$ (0.77), and $SO_4^{2-}$ (0.52). There was a positive correlation between $Ca^{2+}$ and $Mg^{2+}$ (0.78), $Na^+$ (0.35), $K^+$ (0.83), $HCO_3^-$ (0.79), $Cl^-$ (0.68), and $SO_4^{2-}$ (0.23). More positive correlations existed between $Mg^{2+}$ and $K^+$ (0.89), $HCO_3^-$ (0.88), $Cl^-$ (0.34), and $SO4^{2-}$ (0.19); between $Na^+$ and $Cl^-$ (0.65), and $SO_4^{2-}$ (0.64); and between $K^+$ and $HCO_3^-$ (0.75). Figure 8A illustrates the correlations between different variables through a biplot, and the angles between the vectors indicate the status of the correlations between each variable. When two vectors are positively correlated, a small angle between the two variables was observed, and if they meet each other at 90°, they are not likely to have been correlated. On the contrary, a big angle (close to 180°) indicates a negative correlation. The majority of the observed correlations were significant ($p < 0.05$) and indicated the presence of these variables in one or more of the major general components. Therefore, the PCA gave better results under these conditions of correlations, based on the verification of Bartlett's sphericity test where small $p$ values ($p < 0.05$) indicated that the PCA was a convenient method for the current data because the observed $p$ value of the data was lower than 0.001 [73]. The Bartlett's sphericity test results are illustrated in Table S1.

### 3.5.2. Validity of Water for Agricultural Irrigation

A summarization of the PCA output based on SPSS software is presented in Table 1. Generally, seven principal components (PCs) were obtained from PCA. The PCs that had an eigenvalue greater than one were kept, and the rest were removed [46].

**Table 1.** Summarization of the principal component analysis (PCA).

|  | PC1 | PC2 | PC3 | PC1 | PC2 | PC3 |
|---|---|---|---|---|---|---|
| Eigenvalue | 4.99 | 1.84 | 1.30 |  |  |  |
| Variability (%) | 55.42 | 20.49 | 14.49 |  |  |  |
| Cumulative % | 55.42 | 75.91 | 90.39 |  |  |  |
|  | **Factor loadings** | | | **Component Score Coefficient** | | |
| $X_1$ (pH) | 0.46 | 0.01 | −0.70 | 0.092 | 0.017 | 0.539 |
| $X_2$ ($EC_{iw}$) | 0.95 | 0.28 | 0.10 | 0.191 | 0.155 | −0.072 |
| $X_3$ ($Ca^{2+}$) | 0.95 | −0.14 | 0.19 | 0.192 | −0.070 | −0.142 |
| $X_4$ ($Mg^{2+}$) | 0.85 | −0.48 | −0.02 | 0.172 | −0.248 | 0.029 |
| $X_5$ ($Na^+$) | 0.50 | 0.85 | 0.03 | 0.099 | 0.459 | −0.026 |
| $X_6$ ($K^+$) | 0.89 | −0.22 | 0.02 | 0.178 | −0.139 | −0.021 |
| $X_7$ ($HCO_3^-$) | 0.76 | −0.57 | −0.13 | 0.156 | −0.298 | 0.103 |
| $X_8$ ($Cl^-$) | 0.64 | 0.36 | 0.63 | 0.128 | 0.194 | −0.484 |
| $X_9$ ($SO_4^{2-}$) | 0.45 | 0.54 | −0.59 | 0.087 | 0.297 | 0.449 |

Thus, there were three PCs that had an eigenvalue > 1 (Table 1). A scree plot (Figure S2) was used to determine the number of PCs. The three PCs explained about 90.39% of the total variance of the water variable data. PC1 represented about 55.42% of the total variance, whilst 20.49% and 14.49% of the total variance were represented by PC2 and PC3, respectively (Table 1).

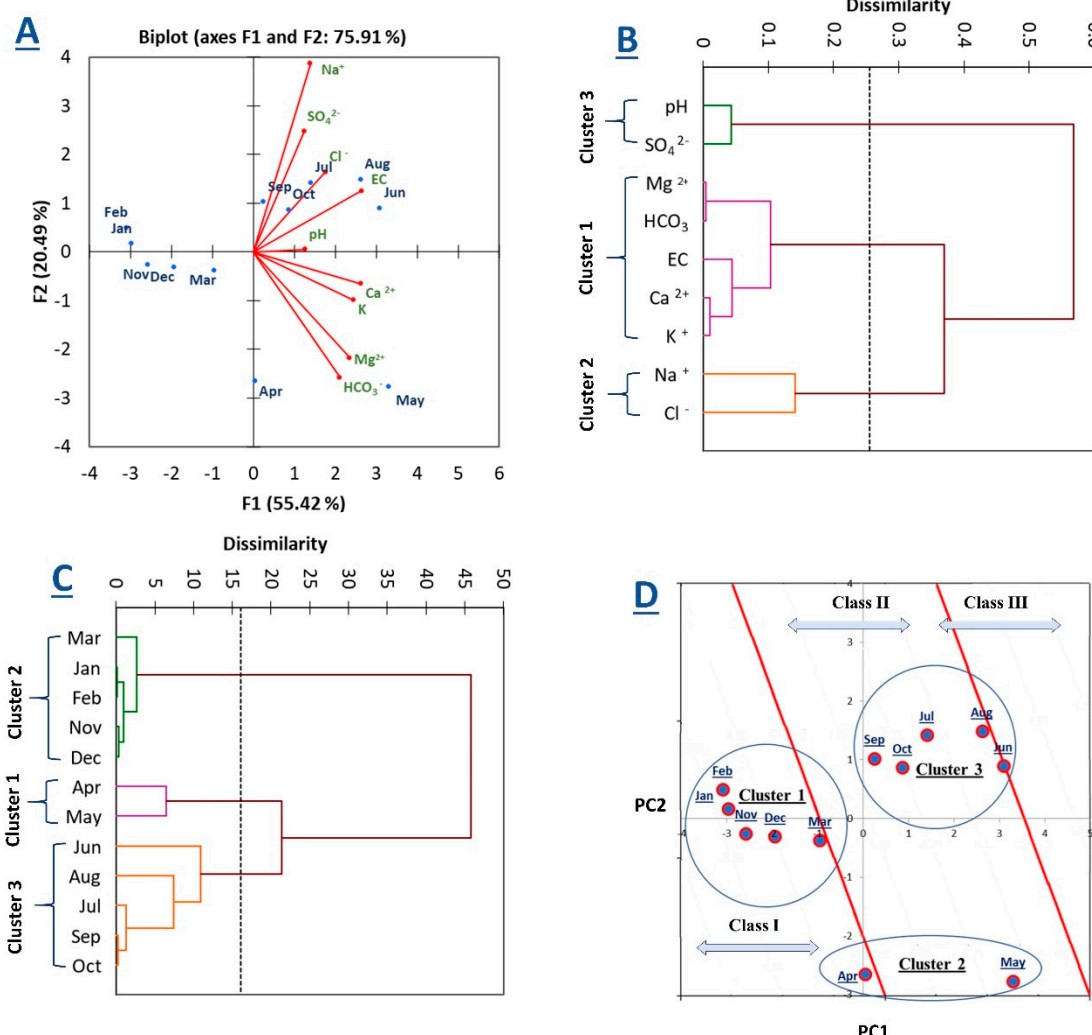

**Figure 8.** Classification of water quality based on the principal component analysis (PCA) and agglomerative hierarchical clustering (AHC) techniques. (**A**) The relationships between water chemical parameters, (**B**) a dendrogram for clustering chemical properties, (**C**) a dendrogram for clustering monthly variation, and (**D**) the final classification of water quality.

Table 1 and Figure S3A show the factor loading of the three PCs, indicating the higher loading and contribution of the corresponding components [77]. Based on the factor loading of the variables in PC1, the contributing descriptors were ECiw, $Ca^{2+}$, $Mg^{2+}$, $K^+$, $HCO_3^-$, and $Cl^-$, where the contributions of the variables were 18.28%, 18.25%, 14.55%, 15.83%, 11.71%, and 8.14%, respectively. PC2 was highly correlated with $Na^+$, where the contribution of the variable was 39.27%. Regarding PC3, the effective contributing descriptors were pH and $SO_4^{2-}$ (negative correlation), where the contributions of the variables were 37.92% and 26.68%, respectively.

A biplot analysis (PC1 vs. PC2) was used to visualize water variables correlations (Figure 8A), and three major groups of water properties were obtained based on the AHC technique (Figure 8B). Therefore, AHC was done to visualize the water variable grouping based on the three PCs that had an eigenvalue greater than 1.

The first group that affected by PC1 was constituted by ECiw, $Ca^{2+}$, $Mg^{2+}$, $K^+$, $HCO_3^-$, and $Cl^-$; therefore, PC1 represents an indicator for salinity and alkalinity hazards because it includes the ECiw, which is normally used to estimate salinity hazard [87]. The ions (i.e., $Ca^{2+}$, $Mg^{2+}$, and $HCO_3^-$) were used to estimate the alkalinity hazard based on the calculations of many criteria such as RSC and RSBC [37,69]. The second group that was affected by PC2 was constituted by $Na^+$; thus,

the PC2 is related to the sodicity hazard where the given hazard indicates the relationship of $Na^+$ with the other cations (i.e., $Ca^{2+}$, $Mg^{2+}$, and $K^+$) that expressed various criteria such as the SSP, the SAR, and the SCAR [69]. The third group was affected by PC3 and constituted by pH and $SO_4^{2-}$; therefore, it may indicate water acidity.

The principal component scores for each month were calculated, and they are presented in Table 2 and Figure S3C, where a higher score indicates lower quality of water. Table 2 and Figure S3C illustrate that all months except January, February, March, November, and December had a high PC1 score (ranged between −2.59 and 3.30); therefore, Bahr Mouise water had some limitations in the months with high PC1 scores. The high PC2 scores were observed in January, February, June, July, August, September, October, November, and December (ranged from −0.31 to 1.49).

**Table 2.** Component score of different months.

| Month | PC1 | PC2 | PC3 | PC Comprehensive Score |
|---|---|---|---|---|
| January | −2.97 | 0.17 | −0.06 | −1.62 |
| February | −3.1 | 0.49 | 0.2 | −1.59 |
| March | −0.96 | −0.37 | −0.42 | −0.67 |
| April | 0.04 | −2.65 | −0.56 | −0.6 |
| May | 3.3 | −2.77 | 0.96 | 1.4 |
| June | 3.09 | 0.9 | −2.7 | 1.5 |
| July | 1.4 | 1.42 | 0.54 | 1.15 |
| August | 2.62 | 1.49 | 2.42 | 2.11 |
| September | 0.25 | 1.02 | −0.57 | 0.27 |
| October | 0.86 | 0.87 | −0.37 | 0.6 |
| November | −2.59 | −0.26 | 0.54 | −1.41 |
| December | −1.95 | −0.31 | 0.01 | −1.14 |

Generally, the limitations (high PCs scores) were observed from May to December. The comprehensive scores were calculated, they are presented in Table 2 and Figure S3C, where August is shown to have had the highest score (2.11) January is shown to have had the lowest score (−1.62). Moreover, Table 1 and Figure S3B,C show the correlations between PC1, PC2, and PC3 with the different variables.

The PC scores could be calculated based on the following equations (Equations (18)–(20)):

$$PC1 = 0.09X_1 + 191X_2 + 0.192X_3 + 0.172X_4 + 0.099X_5 + 0.178X_6 + 0.156X_7 + 0.128X_8 + 0.087X_9 \quad (18)$$

$$PC2 = 0.017X_1 + 0.155X_2 - 0.070X_3 - 0.248X_4 + 0.459X_5 - 0.139X_6 - 0.298X_7 + 0.194X_8 + 0.297X_9 \quad (19)$$

$$PC3 = 0.539X_1 - 0.072X_2 - 0.142X_3 + 0.029X_4 - 0.026X_5 - 0.021X_6 + 0.103X_7 - 0.484X_8 + 0.449X_9 \quad (20)$$

where $X_1$ to $X_9$ represent the measurement of each parameter: $X_1$ is the pH, $X_2$ is the ECiw, $X_3$ is the $Ca^{2+}$, $X_4$ is the $Mg^{2+}$, $X_5$ is the $Na^+$, $X_6$ is the $K^+$, $X_7$ is the $HCO3^-$, $X_8$ is the $Cl^-$, $X_9$ is the $SO4^{2-}$

The comprehensive score can be calculated as shown in Equation (21):

$$PC = 0.5542\ PC1 + 0.2049\ PC2 + 0.1449\ PC3 \quad (21)$$

Table S4 represents the correlation components matrix that indicates that there was no correlation between the components; therefore, each component represents a discrete component.

Lastly, AHC was done to classify the water quality based on PC score results (Figure 8C,D). The class centroids and central objects for each cluster are shown in Table S5. Based on the AHC water quality classification, three water quality classes were obtained; Class 1 was the water quality that was allocated to class I was observed in January, February, March, April, November, and December. The quality of water during the given months did not have salinity, sodicity, and/or alkalinity limitations and could be used for irrigation without any restrictions. Class 2 was water quality that was allocated

to class II that was observed during June, July, August, September, October, and December, and an alkalinity hazard may occur as a result of the continued irrigation during the given months. No months were allocated to class III. If one of the months had class III, the farmers would have to consider proper water management (e.g., providing a good drainage system and calculating the water leaching requirements) to prevent salt accumulation in the soil.

## 4. Discussion

The increasing demand for high-quality water for irrigation in Egypt in the last few decades has led to the use of poor-quality water for irrigation purposes in farmland, while the continuous usage of the low-quality water could cause a decline in crop productivity [88–92]. The current study shows that the water of the Bahr Mouise canal was classified as a moderate salinity hazard; this type of water has low limitations and is suitable for most agriculture crops [69]. Moreover, moderately salt-tolerant plants can be grown in most cases without special management, based on the guidelines for water quality for irrigation presented by the authors of [80]. The water of Bahr Mouise has a low and moderate salinity, but salt accumulation in soil may occur over the long term, and it may cause soil degradation and negatively impact crop productivity [92,93].

Phogat et al. [94] used the traditional criteria in order to evaluate the long-term impact of recycled water irrigation on soil chemical properties and crop yield. Irrigation with recycled water can potentially increase the soil solution salinity, the sodium adsorption ratio, and the exchangeable sodium percentage in the soil. The increased soil salinity reduces the potential almond yield by 12–20% in various soils. At the local farms that are irrigated by the Ethiopian Leyole River, the grown crops such as onion, carrot, potato, and cucumber would be sensitive to the concentration of chemical parameters such as $BO_3^{3-}$, $Na^+$, and the SAR. Higher $Na^+$ and SAR values would lead to soil permeability problems. High SAR levels in irrigation water can cause soil problems such as soil surface crust formation, poor drainage, and poor soil tilth [80,91,95,96]. An excess of $HCO_3^-$ concentration in irrigation water negatively affects the plant uptake and metabolism of nutrients. The excess of Na+ concentration in irrigation water leads to an increase adsorbed exchangeable sodium, which may cause dispersion in soil aggregates, blocking pores and reducing the water infiltration [20]. Soil function is generally threatened by increased food demands, human influence, and its activities (such as the continuous irrigation with low quality water), as well as land use and climate change [97,98]. This may lead to physical and chemical degradation processes and negatively affect soil sustainability [99,100].

Monitoring land use/land cover changes is fundamental for sustainable irrigation water management plans. The authors of [101,102] showed the importance of using the remote sensing techniques in order to detect LCLU changes and their impact on surface water in Nile Delta. Urban sprawl and rapid LCLU changes in the Nile Delta have a negative impact on water irrigation quality by increasing the concentration of contaminants [103,104]. In addition, using agricultural drainage water for irrigation causes degradation to the most fertile soils [6,27,37]. The NDVI assists in the detection of cultivation changes and crop growth during the year, while variations in crop types and growth consequently lead to fluctuations in irrigation water demand, where summer crops need more water compared with winter crops [105]. Moreover, in summer, the temperature and evapotranspiration are higher than other cultivation seasons, so the concentrations of some elements may raise and lead to an increase in the salinity and alkalinity hazards [106,107]. The high air temperatures during the year in arid regions negatively impact crop evapotranspiration, water irrigation requirements, and soil salinity level. Consequently, the plant growth and crop yield are significantly affected [107].

Here, the IWQI was used to assess the suitability of water bodies for different usage purposes, e.g., irrigation, aquatic life, and drinking. Furthermore, it can provide good information to decision makers in order to make suitable decisions about alternative water usage and conserving water bodies [108]. El Shemy et al. [109] applied the Canadian Council of Ministers of the Environment Water Quality Index (CCME-WQI) according to the Egyptian water quality standards for surface waterways Law 48/1982—Article No. 60 [110,111] to assess the overall water quality status of Lake Nubia (southern

Egypt). Seven water quality parameters were integrated into the CCME-WQI and followed the Egyptian water quality standards (e.g., pH (7–8.5), dissolved oxygen (>5 mg L$^{-1}$), nitrate–nitrite (>45 mg L$^{-1}$), total phosphorus (>1 mg L$^{-1}$), total dissolved solids (>500 mg L$^{-1}$), ammonium (>0.5 mg L$^{-1}$), and fecal coliforms (<2000 N/100 mL)); the obtained results indicated that the water quality ranged from 92% to 100% (good and excellent) according to the CCME-WQI.

Several studies around the world have applied PCA to evaluate and classify water quality for irrigation purposes, including China [112], Turkey [113], Nigeria [114], and Brazil [115]. China applied PCA to the Tongjiyan irrigation area, and the given study indicated that the PCA classification had similar results with other methods such as fuzzy evaluation, the Nemerow index, and improved Nemerow index methods. The authors of [116] identified four principal components in order to evaluate overall water quality. On the other hand, the seasonal and temporal variations of irrigation water quality in the Ajakanga area, Ibadan, Nigeria were examined by Ganiyu et al. [117] using PCA, and they reported that 95.7% (dry seasons) and 88.7% (wet seasons) of the total variance of the data set were represented by five principal components. The seasonal alterations in water quality were related to the weathering process, mineral dissolution, groundwater–rock interaction, and anthropogenic activities. Thus, the PCA represents a new approach to achieving the sustainable management of water resources [4,5]. In the same trend, cluster analysis (CA) and PCA were applied to analyzing 36 physicochemical parameters of water samples that were collected from a polluted lake in order to group five sampling sites into three clusters of similar water quality characteristics. The PCA method can be used for identifying physiochemical parameter correlations and the factors responsible for water quality variations [56,118].

As in many studies of water irrigation quality assessments, this study had some limitations. For example, the applied methods and the developed simplified water quality assessment were applied to an irrigation canal with a good water quality; therefore, further studies should be conducted in irrigation and drainage canals with diverse water qualities in order to justify and observe the variations between the applied methods. Nevertheless, the current research showed that temporal analysis is more important than spatial analysis for the same irrigation canal and that more changes were observed in water quality in different studied months than from one site to another.

## 5. Conclusions

The proper management of irrigation water depends on understanding irrigation water quality, as it assists in determining suitable crops and the potential agriculture uses. Irrigation water quality is affected by human activities, agricultural practices, and environmental conditions (e.g., urbanization, use of agrochemicals, and climate conditions). The irrigation water quality of the Bahr Mouise canal (El-Sharkia Governorate, Egypt) at six site locations over the twelve months of 2019 were assessed by using multivariate analyses (i.e., IWQI), as well as by developing a new classification method based on analyzing water chemical composition using the PCA and AHC techniques. The PCs of PCA explained 90.39% of the total variance of water data. The quality of Bahr Mouise canal water was decreased to class II in summer (June, July, August, September, and October), while the highest quality (class I) was observed during the winter (January February, March, April, November, and December).

The new approach of classification based on the PCA water quality as a statistical-based method had similar results to the IWQI method, except for in the assessment of water in September, which was assigned to class II when applying the PCA method but assigned to class I by the IWQI method. The dominant LCLU around the Bahr Mouise canal was agriculture use, and the variations in NDVI values indicated the dissimilarities of crop types and growth stages during the different cultivation seasons. Therefore, the variations in crop types and the status of growth stage were related to crop water requirements and irrigation water quality. Generally, the Bahr Mouise canal was found to have a good water quality for irrigation, but the decline in water quality during the summer indicated that continued irrigation (over the long term) during this season may cause salinization problems and, consequently, negatively impact crop productivity; therefore, the suitable planning of irrigation

water management is highly recommended. Finally, irrigation water represents a principal issue for sustainable agricultural production in Egypt; hence, irrigation water quality must be monitored during the year, and good-quality irrigation water should be used to sustain soils and agricultural productivity.

**Supplementary Materials:** The following are available online at http://www.mdpi.com/2073-4441/12/9/2537/s1, Figure S1: Impact of different land uses on irrigation water quality, Figure S2: The eigenvalue scree plot for determine the number of PCs in the analysis, Figure S3: Component loadings for the three PCs (A), PC scores for the three PCs (B), and PC comprehensive for the months (SX2), Table S1: Results of Kaiser–Meyer–Olkin (KMO) and Bartlett's sphericity tests, Table S2: Annual average of chemical composition analysis of Bahr Mouise canal water, Table S3: Criteria for judging the validity of Bahr Mouise canal water, Table S4: Correlation components matrix; Table S5: The Class centroids and Central objects for each cluster.

**Author Contributions:** Conceptualization, M.K.A.-F., A.A.A., and A.S.E.; introduction, M.K.A.-F. and E.S.M.; methodology, M.K.A.-F. and S.K.A.-E.; results and discussion, M.K.A.-F., S.K.A.-E., and E.S.M.; review and conclusions S.K.A.-E. and E.S.M.; funding acquisition, A.A.A. and M.K.A.-F. All authors have read and agreed to the published version of the manuscript.

**Funding:** The authors would like to declare that the funding from the study was supported by the Soil Science Department, Faculty of Agriculture, Zagazig University and National Authority for Remote Sensing and Space Science (NARSS), Egypt and Research Group Project no. RGP–VPP–275., King Saud University, Saudi Arabia.

**Acknowledgments:** The authors would like to thank the Soil Science Department, Faculty of Agriculture, Zagazig University for funding the laboratory analysis. The authors would like to National Authority for Remote Sensing and Space Sciences for supporting the satellite data and image processing. The authors would like to extend their sincere appreciation to the Deanship of Scientific Research at King Saud University for its funding of this research through the Research Group Project no. RGP–VPP–275.

**Conflicts of Interest:** The authors declare no conflict of interest.

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
