# Peer review of "Multivariate Analysis for Assessing Irrigation Water Quality: A Case Study of the Bahr Mouise Canal, Eastern Nile Delta"

_water, doi:10.3390/w12092537_

Round 1
Reviewer 1 Report
The purpose of study is to develop a simplified water quality evaluation using principal component analysis (PCA) and agglomerative hierarchical clustering technique (AHC) to assess the water quality of Bahr Mouise canal, El-Sharkia Governorate, Egypt during 2019. Results of he study may have important application in irrigation water management. Authors may wish to consider the following in revising their manuscript.
- Please explain why Bahr Mouise canal was selected in the study.
- Please provide information regarding what types of crops which use irrigation water reported in the study.
- Please provide detailed information regarding sampling stations, sampling frequencies and number of samples collected in each station for data used in the analysis.
- Please provide information regarding irrigation water quality standard in Egypt.
- Please provide TDS data for irrigation water.
- Please provide TOC data of irrigation water.
- Please comment on the use of irrigation return water in the study area.
- Please comment on the limitation of methodology used in the study.
- Please comment on the proposed methodology as compared to other methods reported in the literature.
Author Response
The purpose of study is to develop a simplified water quality evaluation using principal component analysis (PCA) and agglomerative hierarchical clustering technique (AHC) to assess the water quality of Bahr Mouise canal, El-Sharkia Governorate, Egypt during 2019. Results of the study may have important application in irrigation water management. Authors may wish to consider the following in revising their manuscript.
RESPONSE: We thank the reviewer for these positive comments, and we have addressed the nine points in the revised version.
- Please explain why Bahr Mouise canal was selected in the study.
RESPONSE: We selected Bahr Mouise canal because it represents the main irrigation canal that located in El-Sharkia Governorate (Egypt), and it provide the irrigation water requirements of 340000 hectares. We have been addressed this point (highlighted in green) in the section 2.1. Study area and sampling.
- Please provide information regarding what types of crops which use irrigation water reported in the study.
RESPONSE: Generally, the agricultural water requirements in El-Sharkia governorate for summer (cultivated from February–May), winter (cultivated from September–November), Nili crops (cultivated from July–August) and fruits are 2.47, 1.22, 0.31, and 0.03 billion m3, respectively. The major Summer crops are maize, rice, sorghum, cotton, sunflower, sesame, sugarcane, soybean, onion and vegetables. While, the major winter crops are beans, wheat, sugar beet, barley, onion, alfalfa, garlic, lupine, and vegetables. The Nili crops are sunflower, maize, rice, sorghum, onion and vegetables. Therefore, we considered this point (highlighted in green) in the section 3.2. Climate and water use
- Please provide detailed information regarding sampling stations, sampling frequencies and number of samples collected in each station for data used in the analysis.
RESPONSE: To assess the water quality of Bahr Mouise, monthly samples (January to December, 2019) were collected from a random six sampling sites at irregular distance between the point and each other, along the central part of the canal. In each sampling site four samples were collected using a portable water sampler, and all water measurements were carried out within 24 hours after sampling. We addressed this point (highlighted in green) the section 2.1. Study area and sampling
- Please provide information regarding irrigation water quality standard in Egypt.
RESPONSE: We have provided studies in the discussion section (highlighted in green) that assessed the water of river Nile water (southern Egypt) based on Egyptian water quality standard, and we have presented the allowable values of each parameters they used according to water quality standard in Egypt.
“Elshemy and Meon, (2011) were applied the Canadian Council of Ministers of the Environment Water Quality Index (CCME-WQI) according to the Egyptian water quality standards for surface waterways (Law 48/1982 – Article No. 60; Heikal et al.2007; MWRI, 2005) to assess the overall water quality status of Lake Nubia (southern Egypt). Seven water quality parameters were integrated in CCME-WQI and followed the Egyptian water quality standards (i.e. pH (7-8.5), dissolved oxygen (> 5 mgl-1), nitrate-nitrite (> 45 mgl-1), total phosphorus (> 1 mgl-1), total dissolved solids (> 500 mgl-1), ammonium (> 0.5 mgl-1), and fecal coliforms (<2000 N/100 mL), the obtained results indicated that the water quality were ranged from 92% to 100% (good and excellent) according to CCME-WQI application”.
- Please provide TDS data for irrigation water.
RESPONSE: We have provided the TDS data for irrigation water in Table S1. (supplementary martial). Also, we have referred to the TDS values in the section 3.3. Validity of Bahr Mouise water based on traditional criteria (highlighted in green).
“Moreover, the ECiw values were converted to total dissolved solids (TDS) values, based on the equation (TSD, mgl-1 = EC, dSm-1 * 640) that expressed by Gupta (1990), and the TDS were ranged between 243 and 352 mgl-1 and with an average 294 mgl-1.”
- Please provide TOC data of irrigation water.
RESPONSE: Unfortunately, we did not measure the total organic carbon (TOC) but previous study was measured the TOC in different sites on river Nile, and the study indicated that the values of TOC were ranged between 1.84 and 4.43 mgl-1. According to Chapman and Kimstach (1996) the surface waters, TOC concentrations are generally less than 10 mg l-1, therefore the TOC it may not a limiting factor for irrigation water in case of river Nile water.
- Please comment on the use of irrigation return water in the study area.
RESPONSE: There is no use of irrigation return water in the study area, but generally the agriculture drainage water of the studied area should be directed to the newly reclaimed areas in the Eastern Nile Delta. We refereed to this in the introduction (highlighted in green) as the studied area located in the Eastern Nile Delta
“According to Shaban (2020) the limited and fixed water resources, and ever-increasing of water demands in Egypt are the main issues in accelerating practice of agricultural drainage water reuse as an alternative resource to fill the gap between water supply and demand. While, the Egyptian national water strategy stated that drainage water reuse practice would fulfil irrigation water demands for newly reclaimed areas in the Eastern and Western Nile Delta regions.”
- Please comment on the limitation of methodology used in the study.
RESPONSE: We have been commented about the limitation of the study in the discussion section (highlighted in green), and in the methodology the tests that we carried out in order to check the adequacy of the used methodology.
“As many studies of water irrigation quality assessments, this study has some limitations. For example,
the applied methods and the develop simplified water quality assessment have been applied in an irrigation canal with a good water quality, therefore further studies should be conducted in irrigation and drainage canals with diverse water qualities in order to justify and observe the variations between the applied methods. Nevertheless, the current research showed, the temporal analysis is more important than the spatial analysis in the same irrigation canal, where more changes were observed in water quality in different studied months than from site to each other”.
Meanwhile, the Kaiser-Meyer Olkin (KMO) and Bartlett's Tests were applied to measure sampling adequacy and to examine the appropriateness of PCA (Hutcheson and Sofroniou, 1999). The results of these tests were presented in Table S1 in the supplementary materials. Bartlett's sphericity and KMO tests (Table S1) indicated that the PCA is an adequate method based on the current water parameters data; where the value of KOM is greater than 0.5, indicating the adequacy to perform the PCA analysis (Huck 2012; Pallant and Manual 2007; Tabachnick et al., 2007). On the other hand, the p value of Bartlett's sphericity test was less than 0.05, thus the current variables are related and suitable (Hutcheson and Sofroniou 1999).
- Please comment on the proposed methodology as compared to other methods reported in the literature.
RESPONSE: In the section 3.3. Validity of Bahr Mouise water based on traditional criteria analysis we added new text that include the ranges of classes in each selected criterion according to the international references, then we detect were the data of Bahr Mouise canal allocated in. Also, we added in the discussion section some previous studies that used the proposed methodology for the assessment of water quality in different countries around the world.
“Serval studies around the world have applied the PCA technique to evaluate and classify the quality water for irrigation proposes, in China (Tao et al., 2016 and Hao et al., 2013), Turkey (Yurtseven, 2017), Nigeria (Okonofua et al., 2019 and Ganiyu et al., 2018), Brazil (Garcia et al., 2017), Bangladesh (Mahmud et al., 2007) and Inner Mongolia (Jiang et al., 2015). Tao et al., (2016) applied the PCA in Tongjiyan irrigation area, China and the given study indicated that the PCA classification had similar results with other methods such as fuzzy evaluation, Nemerow index and improved Nemerow index methods. While, Hao et al., (2013) were identified four principal components in order to evaluate the overall water quality. On the other hand, the seasonal and temporal variations of irrigation water quality in Ajakanga area, Ibadan, Nigeria was examined by Ganiyu et al., (2018) using PCA and reported that 95.7 (during dry) and 88.7% (wet seasons) of the total variance of the data set were represented by five principal components. Whereas, the seasonal alterations in water quality were related with weathering process, mineral dissolution, groundwater–rock interaction, and anthropogenic activities”.
Reviewer 2 Report
Dear Authors,
I had the opportunity to read your submission to Water titled "Multivariate Analysis for assessing irrigation water quality. A case study, Bahr Mouise canal, Eastern Nile Delta", but I am sorry to inform you that your manuscript suffers of serious linguistic flaws that make quite hard a right comprehension of the text.
Since the topic you discussed could be of potential interest for the audience of "Water", I warmly suggest to resubmit your paper after a deep revision of the English by a professional proofreading service.
Author Response
RESPONSE: We thank the reviewer for these comments. We have improved all the manuscript’s sections, and a deep revision of the written English have been done.
Round 2
Reviewer 2 Report
Dear Authors, I'm sorry to inform you that your submission continues to suffer of very serious linguistic flaws.
You declared in your reply that you edited the text for correcting the English but, as evidenced in the attached pdf, the revised version is plenty of grammar and punctuation errors and several phrases are badly constructed making difficult to understand what you meant.
I had to stop at line 55 of the first page due to the huge number of errors: making the linguistic revision is not a duty for scientific reviewers . Please use a professional proofreading service prior to resubmitting your paper and include the related certificate/proof

Author Response
The written English has been improved though the professional proofreading service, please see the attach certificate/proof

Round 3
Reviewer 2 Report
Please find specific comments in the attached pdf

Author Response
Thanks a lot. We addressed all of your useful comments in the attached version. Also, we modified table one and the positions of figures in the text, that we believe makes the results much clearer.
